# A Hybrid Pipeline to Assess Oestrogen Receptor Stained Nuclei in Invasive Breast Cancer

Hammam M. AlGhamdi[1], Maryam Althobiti[2], Talha Qaiser[1], Andrew R. Green[2], Shan E Ahmed Raza[1], Emad A. Rakha[2], and Nasir Rajpoot[1]

[1] University of Warwick, Coventry, UK
{hammam.alghamdi,t.qaiser,shan.raza,n.m.rajpoot}@warwick.ac.uk
[2] University of Nottingham, Nottingham, UK
{maryam.althobiti,andrew.green,emad.rakha}@nottingham.ac.uk

**Abstract.** Oestrogen Receptor (ER) expression status in invasive breast cancer not only determines the use of endocrine therapy but its level of expression also provides critical prognostic and predictive information. Digital pathology opens new avenues for applications of computational algorithms to provide objective and accurate assessment of ER status. In this study, we propose a novel hybrid pipeline that combines deep learning (DL) and relatively inexpensive colour histogram features in order to recognise and assess different cell types, including ER positive (ER+) and negative (ER-) tumour cells. Our pipeline consists of a deep neural network for simultaneous detection and classification (*SimNuc-Net*) of nuclei, followed by unsupervised hierarchical clustering. First, the *SimNuc-Net* classifies ER+ and ER- invasive tumour nuclei and nuclei of other cell types. We then classify all ER+ nuclei into four categories based on staining intensity. We show that the proposed pipeline outperforms the DL only pipeline and other existing techniques.

**Keywords:** Computational Pathology · Nuclear detection and classification · Oestrogen (ER) receptors · Breast cancer.

## 1 Introduction

Oestrogen Receptor (ER) is a powerful prognostic and predictive factor in breast cancer (BC) patients [1]. Approximately 70% of BC patients are ER positive and endocrine therapy is determined primarily as standard systemic treatment based on ER positivity [2]. ER has been consecutively reported in clinical routine practice using Immunohistochemistry (IHC) in BC tissue and is assessed by pathologists visually estimating the distribution of ER expression across all the tumour cells.

In clinical practice, ER expression is evaluated at low magnification, e.g., at the Whole Slide Image (WSI) level. Every *clone*, a group of ER stained nuclei, is visually examined to assess the degree and the percentage of cells stained [3], sometimes leading to the lack of reproducibility due to subjectivity as it relies

on visual analysis by the pathologists. Therefore, alternative automated techniques need to be explored that can make the evaluation of ER expression more reproducible and in more detail at the cellular level.

Recent studies proposed techniques and automated tools to evaluate the ER expression in BC tissue. Trahearn *et al.* [4] presented an automatic method for scoring ER, utilising IHC and Haematoxylin and Eosin (H&E) images. Their method detects tumour areas from H&E images, and then categorises ER+ nuclei that present in tumour areas into different categories based on staining intensity. The process of categorising ER+ is based on manual thresholding. However, applying such an approach would be challenging as it requires frequent adjustments of thresholds due to the variability among different histopathological images [5]. Using only H&E images, Lu *et al.* [6] proposed an approach that predicts survival, by extracting hand-crafted features of nuclear shape and orientation. Bucheli *et al.* [7] showed that information extracted from tubule nuclei is able to calculate an automatic risk score that is correlated with the risk category from Oncotype DX test, which in part measures the aggressiveness of ER expression. These methods [6,7], however, do not exploit the wealth of information in IHC images.

In recent years, deep learning (DL) networks have been shown to perform significantly well in the field of Computational Pathology, see for example [8]. Since DL network require training with a large amount of data to achieve good performance [9], it is not always possible to benefit from DL methods in medical applications where providing such huge data is not feasible. An alternative way is to combine DL methods with classical methods (using hand-crafted features) to improve the performance. Several recent studies have shown improvement when combining both types of approaches, for example [10–12].

In this paper, we propose a novel hybrid pipeline that addresses the above limitations. First, we propose a customised DL network, *SimNuc-Net*, to detect and classify 4 types of cell nuclei, including positive tumour (ER+), negative "Unstained" tumour (ER-) and other types of nuclei. Second, we further classify the ER+ tumour cells into 4 categories based on staining intensity which we refer as very weak (vw), weak (w), moderate (m), and strong (s) using simple unsupervised hierarchical clustering on colour histogram features. Our main contributions of this paper are as follows:

1. We present a novel pipeline assessing the degree of ER+ expression in BC histology images.
2. We propose a customised deep learning model that simultaneously detects and classifies various types of nuclei in BC.
3. We combine the deep model with a simple clustering approach using colour histogram features and show that the proposed hybrid approach performs better than the DL model alone.

We evaluated the proposed pipeline by comparing its results with existing techniques. Our results show that the proposed pipeline significantly improves the classification performance, with overall $f_1$-*score* of 0.87. The second best performance was obtained when combining DL with the hand-crafted method

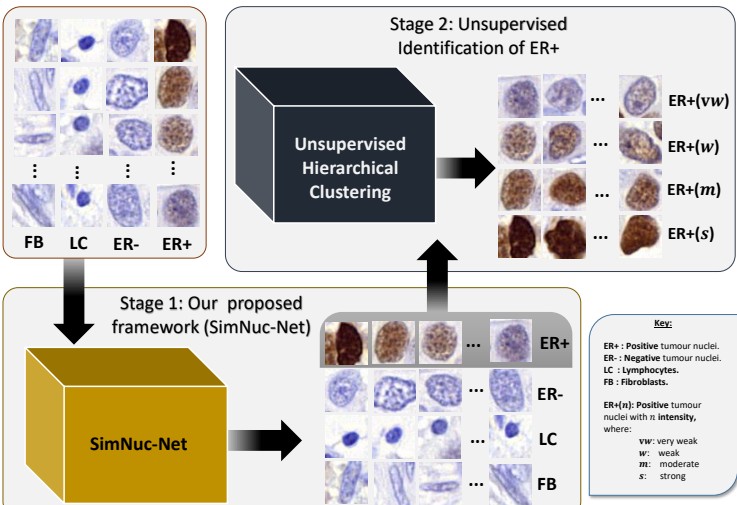

**Fig. 1.** An overview of our hybrid pipeline

used in [4]. These results suggest that DL networks would be able to perform better when combining them with the classical hand-crafted methods for ER cell classification.

This paper is organised as follows. In Section 2, we describe the material and our methodology, followed by a discussion of our results in Section 3. We conclude the paper in Section 4.

## 2    Materials and Methods

### 2.1    Materials

For this study, the formalin fixed paraffin-wax-embedded (FFPE) blocks have been retrieved from Nottingham Health Science Biobank. We used the histopathology database in Nottingham City Hospital to identify the patients with primary invasive breast cancer with known ER status. The data has been reviewed independently, and the annotations have been done by MA.

### 2.2    The Proposed Methodology

Our proposed pipeline contains two stages, as shown in Fig. 1. In the first stage, we propose a customised DL network, *SimNuc-Net* that simultaneously performs nuclear detection and classification. *SimNuc-Net* identifies 4 different types of nuclei, including lymphocytes (LC), fibroblasts (FB) and negatively stained tumour (ER-), and positively stained tumour (ER+). For the second stage, we further classify ER+ into 4 different classes based on staining intensity

by using unsupervised hierarchical clustering. Those ER+ nuclei are very weak (vw), weak (w), moderate (m), and strong (s).

**Stage 1: Simultaneous Cell Detection and Classification Network (*SimNuc-Net*).** We propose a multi-task learning framework (*SimNuc-Net*) that simultaneously performs cell detection and classification. Our network consists of two heads: 1) *classifier:* classifying the type of nuclei, 2) *regressor:* predicting its centre point. We illustrate the proposed architecture of *SimNuc-Net* in Fig. 2.

For a given patch $x_n$, where $n \in (1, 2, ...., N)$ is the number of patches, the model predicts the class labels $y_n^p$, where $p$ is the number of predicted classes. Besides, the proposed framework also predicts the centre location $l(r, c)$ of each $x_n$.

The framework processes $x_n$ through a stack of convolutional layers (CL), followed by a *ReLU* activation function. In order to preserve spatial resolution of $x_n$, we used convolutional kernels of relatively small receptive fields, including kernels of $2 \times 2$ and $3 \times 3$, as well as adjusting a stride of 1 pixel for all CLs.

A spatial dropout layer is followed by *softmax* layer in the classifier head predicting the belonging of $x_n$ into $y_n^p$, where $p \in (1, 2, 3, 4)$, which is the predicted classes, including ER+, ER-, FB, and LC. For the regressor head, the *sigmoid* layer is responsible for predicting the centre location $l(r, c)$ of the nucleus for each $x_n$.

During training, the weights of our proposed framework are jointly optimised in order to enable the model to unify the cell detection and classification tasks. There are three different types of weights $w \in (w_c, w_r, w_s)$, where $w_c$ are the weights in the classifier head, $w_r$ are the weights in the regressor head, and $w_s$ are the weights in the main stream (encoder in Fig. 2). The loss function $L$ is computed as follows:

$$L = L_c(y, \hat{y}) + L_r(l(r, c), \hat{l}(r, c)) \tag{1}$$

Where $L_c(y, \hat{y})$ computes the *log* loss between the predicted classes $\hat{y}$ and true classes $y$. On the other hand, $L_r(l(r, c), \hat{l}(r, c))$ is responsible for calculating the loss between the predicted location $\hat{l}(r, c)$ and the true one $l(r, c)$.

**Stage 2: Unsupervised Identification of Stained Tumour Nuclei.** Upon classification of ER+ cells, the next task is to cluster the ER+ cells into four categories. One straightforward way of handling this problem is to train the *SimNuc-Net* with 7 classes instead of 4 classes. Given the intra-class heterogeneity lies within the ER+ cells, the effective training of *SimNuc-Net* requires handful number of training images from all 4 ER+ categories. In a clinical practice, IHC scoring is generally performed on the WSI level and therefore the inevitable fact is that attaining large-scale ground-truth for each ER+ category is a strenuous task for pathologists. To elevate this problem, we leverage unsupervised hierarchical clustering to separately identify ER+ cells into very weak, weak, moderate and

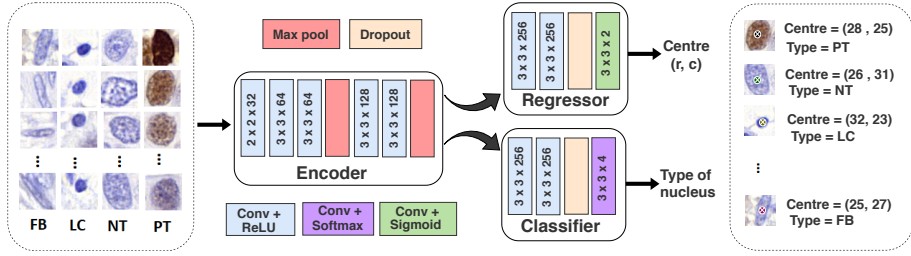

**Fig. 2.** The architecture of the proposed *SimNuc-Net*

strong categories. In Section 3, we show that the performance of categorising ER+ using hierarchical clustering is improved.

We prefer to use agglomerative hierarchical clustering due to its reproducibility. Other clustering techniques, such as k-means and k-Mediods, require initial points to iteratively form the clusters, and in most application these points are chosen randomly or manually [13].

The data used in this stage is intensity-based features, which is the distribution of colour channels (Red, Green, and Blue or RGB) of every given patch. In addition to that, we use the histogram of the DAB channel from stain deconvolutions method used in [4].

Given patch input is $x_n$, we extract $s_n$, which is the feature set of $x_n$. The method starts with joining every pair of $s_n$ that have the minimal *Spearman* distance method, into one *cluster*, which is $c_m$. Next, the method iteratively merges small pairs of $c$ that have minimal distance into larger ones. Eventually, it builds the hierarchy of clusters, producing a binary tree, which can divide the dataset into four of clusters.

We first build the hierarchy of clusters using the same training dataset that were used in *Stage 1*, some patches are shown in Fig. 3. To find out which cluster belongs to which class (type of nuclei), we measure the centroid of each cluster. During testing phase, we measure the distance between every patch in the testing dataset and the centroid of clusters, that are calculated during training. The smallest distance contributes to the category of each ER+.

## 3 Experiments and Results

### 3.1 Experimental setup and Datasets

*Datasets.* We performed experiment on 17 WSIs for patients with early stage ER+ BC. The cohort is randomly divided, at the WSI level, into three datasets: (i) 8 for training, (ii) 4 for validation, and (iii) 5 for testing. We cropped patches of size $51 \times 51$ at $40\times$ magnification for each of the annotated nucleus. Examples of extracted patches are shown in Fig. 3. Note that, based on our experiments, images of different size led to roughly similar performance ($\pm 15\%$), therefore we picked the average patch size.

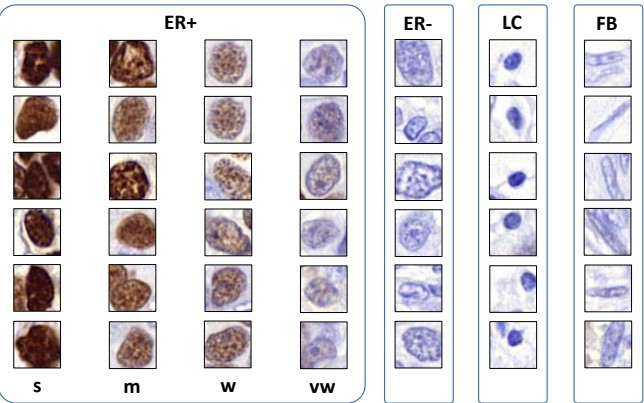

**Fig. 3.** Examples of different types of nuclei in our dataset

*Ground-truth.* The number of annotated nuclei reached up to 3067. These annotations were annotated by MA and are roughly balanced among different nuclear types (different classes). The difference between the smallest and largest class is approximately 70%.

*Data Augmentation.* For each nucleus, we extracted 4 patches where the location of nucleus within each patch varies, therefore the network becomes robust against the variations in the position of nuclei. Additionally, different augmentation methods were applied randomly during training, i.e., (flipping vertically/horizontally or rotating image ($90°, 180°$, or $270°$).

### 3.2    Evaluation and Comparison

Table 1 presents the results of performance metrics using our proposed pipeline, in addition to applying some existing methods. Three performance metrics (precision, recall, and $f_1$-*score*) are presented in the rows for each method. The columns show all 7 types of nuclei, including different ER+ intensity-based nuclear types. Additionally, we show the average of performance metrics for the 4 ER+ types of nuclei, in **ER+ avg** column, as well as the average of performance metrics for all 7 classes in **overall avg** column.

*Comparative analysis.* We applied other existing techniques that are recently proposed for similar problem, i.e., patients with early stage ER+ in BC. Table 1 shows the results of four different methods where we compare the proposed pipeline (method No. 4) with other methods (No. 1, 2, & 3). First, ConvNet [14] is applied, followed by applying only our *SimNuc-Net* detecting all 7 types of nuclei.

**Table 1.** The performance of classifying all the seven classes using different methods.

| No. | Method | Metrics | ER+ | | | | ER+ avg | ER- | LC | FB | Overall avg |
|---|---|---|---|---|---|---|---|---|---|---|---|
| | | | $s$ | $m$ | $w$ | $vw$ | | | | | |
| 1 | ConvNet [14] | precision | 0.63 | 0.38 | 0.38 | 0.60 | **0.50** | 0.57 | 0.46 | 0.37 | **0.49** |
| | | recall | 0.17 | 0.81 | 0.09 | 0.54 | **0.40** | 0.57 | 0.74 | 0.23 | **0.47** |
| | | $f_1$-score | 0.27 | 0.51 | 0.15 | 0.57 | **0.38** | 0.57 | 0.57 | 0.29 | **0.43** |
| 2 | SimNuc-Net | precision | 0.99 | 0.80 | 0.57 | 0.86 | **0.80** | 0.83 | 0.99 | 0.78 | **0.83** |
| | | recall | 0.73 | 0.85 | 0.95 | 0.54 | **0.77** | 0.78 | 0.83 | 0.91 | **0.80** |
| | | $f_1$-score | 0.84 | 0.83 | 0.71 | 0.66 | **0.76** | 0.81 | 0.91 | 0.84 | **0.80** |
| 3 | SimNuc-Net + Thresholding [4] | precision | 0.84 | 0.56 | 0.91 | 0.90 | **0.74** | 0.94 | 0.94 | 0.86 | **0.86** |
| | | recall | 0.52 | 0.81 | 0.95 | 0.99 | **0.75** | 0.82 | 0.97 | 0.94 | **0.84** |
| | | $f_1$-score | 0.65 | 0.66 | 0.93 | 0.94 | **0.75** | 0.88 | 0.95 | 0.89 | **0.84** |
| 4 | SimNuc-Net + clustering | precision | 0.99 | 0.69 | 0.85 | 0.88 | **0.86** | 0.94 | 0.94 | 0.86 | **0.89** |
| | | recall | 0.63 | 0.88 | 0.98 | 0.98 | **0.82** | 0.82 | 0.97 | 0.94 | **0.87** |
| | | $f_1$-score | 0.77 | 0.77 | 0.91 | 0.93 | **0.84** | 0.88 | 0.95 | 0.89 | **0.87** |

The second best performance, after out proposed pipeline (method No. 4), is obtained when we also combine DL network with hand-crafted (method No. 3). The same proposed pipeline is applied; however, the second stage is substituted with the intensity *thresholding* method used in [4]. The overall average of $f_1$-*score* is 0.84, which is a considerable improvement. However, these results were obtained after a number of trials of threshold tuning. Thus, unsupervised approach, like clustering, would be able to self-tune the appropriate thresholds. Overall, it clearly shows that our proposed pipeline outperforms other methods.

*Visualising intensity-based features.* To examine whether intensity-based features are able to discriminate between our 4 ER+ nuclear types, t-SNE algorithm is used to visualise these features. It is inferred from Fig. 4 that intensity features are discriminative enough to separate between different classes. As expected, different ER+ nuclei lie on a continuum where very weak nuclei (red dots in Fig. 4) are found next to weak (green dots), moderate (light blue dots) and finally strong (purple dots) nuclei.

## 4   Conclusions and Future Work

Evaluation of ER expression is an important task due to the fact that it is an essential prognostic and predictive factor in BC patients. In a clinical routine practice, ER expression is evaluated by IHC staining of BC tissue followed by a visual evaluation. However, this evaluation is not reproducible and subjective to the experience of pathologists. Therefore, proposing automatic techniques that would ease the process of ER expression evaluation, would not only be beneficial to pathologists, but also would support treatment decision, and eventually patients' lives.

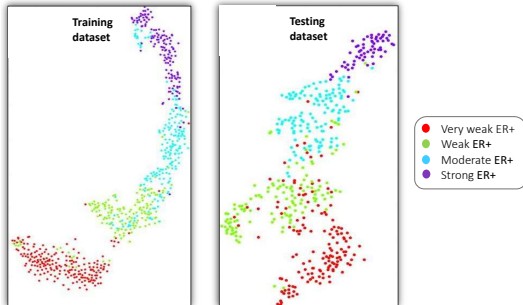

**Fig. 4.** Visualisation of intensity-based features for training & testing data using t-SNE

DL methods require a large amount of annotated data, which is challenging in medical applications. Alternatively, combining DL with classical handcrafted methods would overcome this challenge. Therefore, in this paper, we propose a hybrid pipeline that first classifies ER+ from other types of nuclei. Next, ER+ nuclei are further classified into 4 categories based on their intensity, using a hierarchical clustering method. We find that our proposed pipeline achieves better performance compared to other techniques.

We intend to use the proposed pipeline on WSIs and that may require neighborhood ensembling approach [15] to scale patch level nuclei detection results to WSI level. To do so, we plan to extract overlapped patches from WSIs to find the location and types of different nuclei. This information will assist in our further analysis, such as studying the spacial distribution of ER+. We also aim to use a larger cohort of WSIs, and maybe data from an external institution.

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
