# OpenReview forum: "A Hybrid Pipeline to Assess Oestrogen Receptor Stained Nuclei in Invasive Breast Cancer"
_MICCAI.org/2019/Workshop/COMPAY — COMPAY 2019_

### Official Review · AnonReviewer4 · 2019-08-15

**Rating:** 7
**Confidence:** 3

**Review:**


This paper proposes a two-step approach to classification of ER-positive breast cancer tumor cells. First, a CNN model is used to classify ER+ tumor cells versus ER-, lymphocytes and fibroblasts. Then, classified ER+ cells are collected and a feature vector is extracted from each patch, containing information on RGB colors and DAB after color deconvolution. Authors show that the propose combination of CNN + clustering outperform CNN alone (previous work form the same lab) and CNN + thresholding.
The paper is well written and the presented method seems effective.

Comments:

1.	In this paper, patches containing cells are provided. How are initial those cells cropped, and how would this procedure be applied to whole-slide images in practice?
2.	Is clustering applied to the entire dataset or at WSI level? This should be clarified and might be relevant for clinical application, as cells in the same cluster may not have a comparable expression, because ER stain is known to have variation even within the same pathology lab. Therefore, authors should clarify how they envision using the output of the cluster in practice.
3.	The center of the cell is predicted, but how is it used?
4.	Are there any publicly available datasets on this topic? It would be good to see how this method compares with others on a public dataset.

---

### Official Review · AnonReviewer5 · 2019-08-15

**Rating:** 4
**Confidence:** 5

**Review:**

Summary:
The authors proposed a 2-stage pipeline to classify nuclei in IHC-stained histology images (first stage: detecting and classifying nuclei in 4 categories (ER+, ER-, LC, FB) with a deep learning model, second stage: classifying ER+ detected nuclei in 4 sub-categories with an intensity-based clustering approach).
The authors presented a comparative analysis of different models and reported their classification performances.

Strengths of the paper:
- The paper addresses an important problem.
- The paper proposes an interesting comparison including a classical classification method.
- Interesting study of a 7-class classification problem, split in 2 stages.
- Relevant references.

Weaknesses of the paper:
- The authors claim that their study suggests that DL networks would better perform when combined with hand-crafted features (page 2): this is a strong statement, I think a larger-scale and more in-depth analysis should be done to validate this statement.
- The argument justifying the need for a 2-stage pipeline with hand-crafted features to overcome a lack of annotations to properly train a DL model is not very convincing: other methods addressing this problem should also be compared (strong regularization/augmentation, semi-supervised learning).
Decisions made for the design of the method are unclear:
- What is the purpose of the Regressor? Is the detection component of the model used at test time? What is the detection performance? How does the model would handle non-nucleus detections?
- Is the model evaluated only on pre-selected patches or on larger images? How does this correspond to a real application scenario?
- Some information about the dataset is missing: how many annotations are represented in each dataset split for each class?
- Important details about the training procedure are missing: how are the classes balanced during training? Were ConvNet and SimNuc-Net trained with batches balanced on the 7 classes? if not, this could explain the lower performances of these models. What was the batch size? optimizer? learning rate? Was Batch Normalization used?
Other elements for the experiment/result section would have been expected:
- Experiments should be repeated with random initialization seeds so as to report some variance on the performances.
- No examples of the mis-classification cases were shown/analyzed.

Writing quality:
- several typos/inconsistencies were identified, for example: "the field of Computational pathology field" in page 2 ("field" is repeated); "we performed experiment on 18 WSIs [...]  8 training, 4 validation, 5 testing" in pages 5-6 (does not sum up); "rotating image 90, 180, 90" in page 6 (repeated 90); "due to the fact the it an" in page 7.

Conclusion:
The authors proposed an interesting study on a relevant problem but I would recommend rejecting the paper since many important components are missing to ensure that the comparative analysis was fair and reproducible.
Further analysis, comparisons and additional care to the method/experiment/result description would have been expected to make the paper suitable for COMPAY.

---

### Official Review · AnonReviewer1 · 2019-08-15
**Solid work and clear presentation**

**Rating:** 9
**Confidence:** 4

**Review:**

The paper presented a novel hybrid pipeline composed of a deep learning model to detect and classify ER+ tumor cells, ER- tumor cells, lymphocytes and fibroblasts, and unsupervised hierarchical clustering to further categorize the ER+ tumor cells into four different intensity groups. This novel hybrid model was demonstrated to outperform the previously techniques proposed to address a similar problem.

Overall, the concept presented in the paper is novel and interesting, and justify for publication with some points addressed. The followings are my comments:
(1): The SimNuc-Net was trained with four nuclei categories consisting of ER+ tumor cell, ER- tumor cell, lymphocytes, and fibroblasts nuclei. However, considering the extracted 51 x 51 patch from the WSI in the testing set might not have a nucleus in it, it makes more sense to me to extend four nuclei categories into five categories by adding a group of no-nuclei in the training process. Or maybe the authors employed an alternative approach to address this problem, which, in this case, should be mentioned in the paper.

(2) As stated in the paper, “variability might exist among different histopathological images”. Considering the variation might be even wider between the slides produced from different labs, the robustness of the model presented in this paper could be compellably demonstrated if it is able to yield comparable performance on the WSIs from an external institution other than the one from which the training set comes.

(3) The way to integrate the classifier and regressor as two heads of the network is interesting. It is important to provide more detailed information about how the loss function of the two heads was calculated. Since overall network loss is the sum of the two loss functions, if the value scales of nuclei classification loss and center calculation loss vary significantly, the model may skew severely to the head with the higher scale.

---

### Decision · Program_Chairs · 2019-08-20

Accept